# Industrial Silicon-Wafer-Wastage-Derived Carbon-Enfolded Si/Si-C/C Nanocomposite Anode Material through Plasma-Assisted Discharge Process for Rechargeable Li-Ion Storage

**DOI:** 10.3390/nano12040659

**Published:** 2022-02-16

**Authors:** Rasu Muruganantham, Chih-Wei Yang, Hong-Jyun Wang, Chia-Hung Huang, Wei-Ren Liu

**Affiliations:** 1Department of Chemical Engineering, R&D Center for Membrane Technology, Center for Circular Economy, Chung Yuan Christian University, 200 Chung Pei Road, Chung Li District, Taoyuan City 32023, Taiwan; rmurugaphd@gmail.com; 2Sino Applied Technology Co., Ltd., Chung Li District, Taoyuan City 320030, Taiwan; tyang89@gmail.com (C.-W.Y.); hongjyunwa@gmail.com (H.-J.W.); 3Department of Electrical Engineering, National University of Tainan, No.33, Sec. 2, Shulin St., West Central District, Tainan City 700, Taiwan; chiahung@mail.mirdc.org.tw; 4Metal Industries Research and Development Centre, Kaohsiung 70101, Taiwan

**Keywords:** silicon, anode, plasma, discharge, Li-ion batteries

## Abstract

Silicon is a promising anode material for high-performance Li-ion batteries as a result of its high theoretical specific capacity and elemental abundance. Currently, the commercial application of the Si-based anode is still restricted by its large volume changes during the lithiation cycles and low electrical conductivity. To address these issues, we demonstrate a facile plasma-assisted discharge process to anchor nano-sized Si particles into methanol with quick quenching. After the subsequent sintering process, we obtained a Si/SiC/C composite (M-Si). The unique structure not only allowed for the electrolyte infiltration to enhance lithium ion diffusion during charge and discharge process, but also buffered the volume expansion of silicon particles to enhance the rate capability and cycle stability. The M-Si cell electrochemical results exposed good Li-ion storage performance compared to that of the bare Si used cell (B-Si). The electrode cell consisting of M-Si exhibited remarkable enhanced cyclic stability and sustained the reversible specific capacity of 563 mAhg^−1^ after 100 cycles, with a coulombic efficiency of 99% at a current density of 0.1C, which is higher than that of the B-Si electrode cell that was used. Hence, the as-prepared Si/SiC/C composite is an efficient anode material for Li-ion battery applications. Moreover, these results indicate that the novel plasma-assisted discharge technique will bring a potential durable methodology to produce novel high-performance electrode materials for future advanced large-scale energy-storage applications.

## 1. Introduction

Nowadays, the technology industry is prosperous, and people’s living habits are based on 3C products; the intelligent functions, such as mobile Internet access, video/audio playback, and cloud information reception, give these electronic products a large electrical storage capacity for power supply [1,2]. As a result, the high energy density, high operating voltage, and long cycle performance of secondary lithium-ion batteries (LIBs) are attracting attention for energy-storage devices [3,4]. At present, the anode material of commercial lithium batteries is mostly graphite. However, the theoretical capacity of graphite is only 372 mAhg^−1^ [5], which cannot satisfy the high capacity required for lithium batteries. Therefore, it is highly desirable to develop improved anode materials, such as lithium and silicon.

Silicon is considered to be a potential anode material for rechargeable LIBs because of its abundant resource, high theoretical capacity of 4200 mAhg^−1^ and low working potential of below 0.5 V (vs. Li^+^/Li), which is ten times higher than that of the graphite-based anode [6,7,8,9,10]. However, during lithiation/delithiation processes of the cell, the Si-based anode suffered from the low intrinsic electronic conductivity and huge volume expansion, namely greater than 300%, which causes the silicon particles to fragment and the electrode structure to disintegrate [11,12,13]. As a result, unstable and thicker solid electrolyte interface (SEI) layers caused rapid capacity fade during the cyclic performance [12,14,15,16]. To overcome these issues, Si was post-modified with different kinds of coatings and structural designs to suppress the expansion of the silicon for high-energy LIBs applications.

Recently, carbon-modified Si with a porous structure has a longer cycle life than bare Si because the carbon matrix around silicon as a nanocomposite promotes the superior electronic conductivity and strong structural stability [17,18,19]. Some studies have been focused on the Si-nanocomposites modified with high electrical conductivity of N-doped graphene, multi-walled carbon nanotube (MWCNT), and so on [20,21,22,23,24,25]. Na et al. [22] reported a Si-MWCNT nanocomposite anode for LIBs, using a non-transferred direct current (DC) arc plasma system. The precursors used were a micro-sized Si powder and a commercial MWCNT. Jeong et al. [26] investigated a hard carbon-coated nano-Si/graphite (HC-nSi/G) composite by using a cost-effective hydrothermal carbonization route. The resultant hard carbon with graphitic C played an important role in buffering the volume expansion of Si during the (de)lithiation process and endorsed to the electronic pathway for fast electron transfer. Yue et al. [27] proposed the N-type Si wafers as a Si precursor and the electrical resistivity of 0.001 Ωcm. Initially, the Si wafers were ball-milled to powder form and some parts mechanically crushed via sand-milling to smaller particles of nano-size. Afterward, both milled Si-powders and graphite were mixed with the weight ratio of 5:95 (Si/G) and used as an anode material for LIBs. The Si/G composite maintained a reversible specific capacity of 420.4 mAhg^−1^ after 100 cycles and showed a high initial coulombic efficiency of 85.9%. Kim et al. [28] reported upcycled Si nanomaterials as an anode for LIBs and delivered a specific capacity of more than 1500 mAh/g. Tzeng et al. [29] proposed a composite of silicon carbide (SiC) and graphitic (resorcinol–formaldehyde (RF)) carbon-coated anode material for LIBs. They observed a reversible specific capacity of 843 mAh/g at the 150th cycle at 0.2C.

In this work, the Si/SiC/C composite material was prepared by using the plasma-assisted discharge process (PAD process), and the resultant material was applied as an anode for Li-ion batteries. To the best of our knowledge, rarely studies have been available on Si/SiC/C composite anode through the PAD process for LIBs applications. The PAD-modified Si (M-Si) and commercial bare nano-Si (B-Si) were systematically compared in regard to their physicochemical and electrochemical measurements, using X-ray diffraction (XRD), scanning electron microscopy (SEM), energy-dispersive X-ray spectroscopy (EDX), X-ray photoelectron spectroscopy (XPS), N_2_ isotherms adsorption/desorption, galvanostatic charge/discharge (C/D) profiles, rate capability, and electrochemical impedance spectrum (EIS) analyses. In addition, the PAD method aids the industrial-level production of high-quality Si-based composite anode for large-scale Li-ion batteries applications.

## 2. Experimental Section

### 2.1. Materials and Synthesis

Figure 1 shows the schematic view of the PAD process for the preparation of Si/SiC/C composite material from Si-wafer. The Si (Si wafer: Purity 7N Si, SK Siltron Ltd., Gumi, Korea) and methanol (Purity 99.9%, Methanex) solution were both filled in a self-made plasma reaction chamber, in which stainless steel was used as an electrode. Silicon wafer was cut into strips with a width of 1 mm and a length of 100 mm as electrodes, and argon gas was injected into the chamber as a protective atmosphere. The plasma discharge was used at 11 kV. During the process, plasma and arc vaporized silicon rods were generated. After the silicon falls into methanol, it forms round silicon spheres due to high thermal condensation, and the outer layer was covered by carbon layer and orderly arranged. The conductive carbon composite of the silicon nanocomposite materials was filtered with steel mesh and separated by centrifugation at 1000 rpm to obtain the silicon nanoparticle mud. Then, the nano-silicon composite (M-Si) was obtained after being dried and sintered at 1100 °C for 2 h under Ar atmosphere. The commercial pristine Si nanoparticles (Si) and Si wafer derived without modification of Si (B-Si) with nano-size and containing SiO_2_ (provided from Zhongning Silicon Industry^®^, Xiamen, China) were used to compare the Li-ion electrochemical performance. The physicochemical characterizations’ details are described in the Appendix A.

### 2.2. Electrode Preparation

Assembly of working (anode) electrode: slurries comprising 70 wt.% active material (bare Si and PAD-modified Si), 15 wt.% of conductive agent (10 wt.% of KS-6 and 5 wt.% of Super P, Timcal^®^, Shanghai, China), and 10 wt.% carboxyl methyl cellulose (CMC) with 5 wt.% of styrene butadiene rubber (SBR) used as a binder were mixed in distilled water that was used as a solvent to form a homogeneous slurry. Afterward, the mixed slurry was coated uniformly on a thin Cu-foil, using the doctor-blade coating method. The final electrode was obtained after being dried at 120 °C, overnight, in a vacuum oven. The thickness of the electrode was maintained at approximately 40 μm. Then the electrodes were cut into discs with a diameter of 1.2 cm, and the active Si-mass stacking was 1.8–2.0 mg/cm^2^.

### 2.3. Fabrication of Lithium-ion Half-Cell

A half-cell was fabricated by using a CR2032 coin-type cell with Li metal as the counter electrode and the prepared electrode (anode) as a working electrode. Celgard 2400 porous polypropylene (Type A/E, P/N 61630; Pall Corporation, Xiamen, China) film as a separator was used between the working and reference electrodes. A 1M lithium hexafluorophosphorus (LiPF_6_) in ethylene carbonate: ethylmethyl carbonate: dimethyl carbonate (1:1:1 wt.%) with 1% of vinylene carbonate (UBIQ Technology, Taoyuan, Taiwan) was used as an electrolyte. The coin-cell assembling process was performed in an Ar-filled glove box (MBraun lab star model), with the O_2_ and H_2_O maintained at less than 1 ppm.

### 2.4. Electrochemical Measurements

The galvanostatic technique was used to analyze the discharge–charge characteristics, using a constant-current and constant-voltage programmable AcuTech battery testing system (Taipei, Taiwan, R.O.C, model 750B) in a potential range of 0.01–2.0 V (vs. Li) at different current rates. An alternating current (AC) impedance measurement (electrochemical workstation model: CHI 6273E) was examined in a range of 0.01 Hz to 100 kHz at an AC voltage of 5 mV. All the electrochemical measurements were conducted at room temperature.

## 3. Results and Discussion

Figure 2a shows TGA curve of M-Si under air with a heating rate of 5 °C/min. The initial negligible weight loss was due the desorbed of water molecules on the surface of M-Si powder. The weight loss from 580 to 700 °C can be ascribed to the decomposition of carbon or oxidation of unreacted carbon [30]. From the TG data, the carbon content in M-Si was determined to be ~21%. After 700 °C, a slight increase of weight was the result of a typical oxidation of silicon/Si-C under high temperature in air.

Figure 2b displays the XRD patterns of commercial silicon nanopowder (B-Si) and as-prepared M-Si powder samples in the 2θ range of 10° to 80°. As shown in the XRD patterns, the typical silicon diffraction peaks are located at 26°, 47°, 56°, 68°, and 76°, respectively, in both the B-Si and M-Si samples. It is corresponding to the miller planes of (111), (220), (311), and (400), respectively. The results indicated that the main phase of M-Si was crystalline Si. Apart from the signal of Si, the minor second phases for M-Si shown in 35°, 36°, and 59° were determined to be α-SiC and β-SiC phases [30]. During such a high sintering temperature of 1100 °C, silicon carbides were formed on the surface of M-Si. In addition, the diminished amorphous-like diffraction peaks were observed at ~26° and ~43° in the M-Si sample instead of B-Si sample, as is confirmed by the contribution of carbon [25]. The diffraction peak shown at 13.9° was contributed from the XRD sample holder.

Figure 3a shows the survey scan XPS spectra of B-Si and M-Si samples. The presences of Si, C, and O were confirmed, and the estimated atomic percentage are shown in Appendix A. It can be seen that the M-Si sample exhibited a higher atomic percentage of C (70.99%) and lower atomic % of O (12.59%), which represented the successfully modification with C on the surface of the Si sample. The carbon content in commercial silicon was determined to be 13.9%. Figure 3b displays the C1s high-resolution XPS, and the corresponding fitted spectra are shown in Appendix A. The binding energies are located around 284.6, 285.8, 282.9, and 287.9 eV, indicating the bonding of sp^2^, sp^3^, Si-C, and CO*_x_* on the prepared samples [31]. The M-Si sample exposed 4.37, 5.51, 65.74, and 5.21% of Si-C, CO*_x_*, sp^2^, and sp^3^ types of carbon, respectively. Figure 3c,d display the high-resolution XPS of Si 2p for M-Si and B-Si, respectively. The Si 2p spectra exhibited two individual peaks at binding energies of 98–101 eV for Si-Si bonding and 100.2–10.1 eV for Si-O bonding [32]. As shown in Figure 3d, the B-Si was analyzed by using the high-resolution Si 2p spectrum, in which the ratio of SiO*_x_* bonding was higher than that of Si-Si bonding. The serious oxidation was due to the nature of nano-silicon when exposed in atmosphere. For the M-Si sample, interestingly, the ratio of Si-Si bonding was much higher than that of SiO*_x_* bonding. This means that the Si-C layer on the surface of M-Si could inhibit the oxidation of silicon in atmosphere [32]. The oxygen content of B-Si and M-Si was determined to be 51.28 and 12.59 mol.%, respectively. In addition, the Si-C bonding located at ~102 eV was obviously shown in Figure 3c. The Si and O binding energies are shifted to lower energy in the M-Si sample (Appendix A), which can be ascribed to the oxygen elimination and bonding of Si-C, respectively. The M-Si and B-Si samples’ oxygen high-resolution deconvoluted XPS is shown in Figure 3e,f. The deconvoluted O 1s spectra confirmed the Si-O-Si, C-O-C, and SiO_2_ bonding on the surface of the prepared sample.

Figure 4a,b illustrate the nitrogen adsorption/desorption isotherms for the prediction of specific surface area, pore size distribution, and pore volume of B-Si and M-Si, respectively. The Brunauer–Emmett–Teller (BET) method provided the specific surface area of B-Si and M-Si power at 53.37 and 55.46 m^2^ g^−1^, respectively. The BJH pore size distributions of B-Si and M-Si are shown in the insets of Figure 4a,b, respectively. The average pore size and cumulative pore volume exposed 31.75 nm and 0.491 cm^3^/g for B-Si, and 27.34 nm and 0.539 cm^3^/g for M-Si samples, respectively. However, it can be visibly seen that the M-Si exhibits an ultra-nano-size pore of about 2.6 nm more than B-Si (inset in Figure 4a,b, and this will help to improve the Li-ion storage performance.

Figure 5a,b show the FE-SEM images to examine the morphology of the B-Si sample. The high-magnification image showed almost homogenous spherical particles. Furthermore, to reveal the particle morphology, HR-TEM was carried out, and the resultant images are shown in Figure 5c–j. It can be clearly seen in the particles within 50 nm of size the presence of amorphous types of SiO particles and crystalline Si nanoparticles. Figure 5f shows the SAED pattern of the amorphous with agglomerated crystalline Si region. Figure 5j shows the SAED pattern of the clear Si crystalline particle region, respectively. The observed SAED showed the single crystalline of Si nanoparticles. The lattice and existence of SAED spots are in agreement with the XRD analysis. Figure 6 shows the M-Si sample morphological observations. Figure 6a,b show the FE-SEM images showing the similar morphology of bare B-Si sample. Figure 6c–f display the HR-TEM images, and the particles’ morphology showed C distributed on the surface. The particle size is around 20 nm. Furthermore, the high-magnification HR-TEM images showed a clear view of Si lattice with wrapping and enfolding of graphitic and amorphous-like carbon (Figure 6e,f). The graphitic-like carbon was wrapped with 3.58 nm of thickness on the Si-nanoparticles. The resultant nature of C with Si nanocomposite will promote the electrochemical performance during the (de)lithiation process. Figure 7a,b show the B-Si and M-Si samples HR-TEM coupled with EDX analysis. The M-Si sample mapping images clearly showed the lower amount of O distributions with higher concentration of C content on the surface of the sample. Moreover, it can be obviously seen that the Si and O elements are homogeneously distributed in the B-Si sample. The EDX spectrum view confirmed the presence of elements and their atomic % (Appendix A). The existence of the C layer can shield Si/SiO nanoparticles from directly making contact with the electrolyte solution and decrease the side reactions on the electrode surface during the lithiation/delithiation process.

The electrochemical performance of the bare and modified Si composites anode was examined by galvanostatic charge/discharge test within the potential range of 0.01–2.0 V (*vs*. Li/Li^+^). Figure 8a,b show the discharge–charge curves of bare and modified Si electrodes at initial three cycles tested at a current rate of 0.1C. The discrete plateau profile of the initial discharge cycle was observed below 0.1 V, which can be attributed to the Si-lithiation process [33]. The Si–wafer-wastage-derived bare Si (B-Si) electrode cell exhibited initial discharge/charge capacities of 2991 and 2224 mAhg^−1^, respectively, with an initial coulombic efficiency of 74.36%. The Si-wafer with carbon-modified Si (M-Si) electrode cell showed higher initial discharge/charge capacities of 2853/2231 mAhg^−1^ respectively, with an initial coulombic efficiency of 78.20%. The observed samples’ initial coulombic efficiencies are higher than those of the reported Si-anodes [33,34]. The third-cycle reversible discharge capacities of B-Si and M-Si electrode cells exposed 1147 and 2299 mAhg^−1^ with the corresponding coulombic efficiencies of 78.72 and 92.39%, respectively. During the initial cycle, irreversible capacity loss can be ascribed to the formation of solid electrolyte interface layers and irrevocable lithiation of SiO*_x_* layer and carbon layer into the carbon modified sample [15,35]. The prolonged cyclic capacity test is shown in Figure 8c. The M-Si electrode cell exposed the reversible discharge/charge capacities of 563/557 mAhg^−1^ at the rate of 0.1C over 100 and 250 cycles, and this is higher than that of B-Si used cell (383/374 mAhg^−1^). Moreover, it can be seen that, for the pure Si nanoparticle, a reversible capacity of 100 mAhg^−1^ was exposed over 30 cycles, and the initial coulombic efficiency showed 68%, which was difficult to use for commercialization. The corresponding cycles’ coulombic efficiencies are shown in Figure 8d, and the results showed 100% coulombic efficiency at the end of 250 cycles of both samples. As shown in the cyclic test, the as-prepared cells consisting of M-Si showed gradual capacity fading over 50 cycles, as can be suggested from the slowly infiltration of the electrolyte into the electrode porous, continuous formation of further SEI through the improved specific surface area contained with amorphous carbons, and gradual activation process. The similar gradual capacity fading was observed on reported Si/C and Si/C/SiC composites as anode materials for Li-ion batteries [29,36,37,38,39,40].

The rate performance of the synthesized composites was characterized from 0.2C to 5C. Figure 9a shows each rate cycles end (20th) of the reversible discharge/charge curves of B-Si electrode cell. The B-Si electrode cell’s 20th-cycle charge capacities were 489, 410, 343, 269, 198, and 177 mAhg^−1^ at 0.2, 0.5, 1, 2, 3, and 5 C, respectively. Figure 9b illustrates the B-Si electrode cell corresponding rate cyclic test. Moreover, the M-Si electrode cell’s 20th-cycle charge capacities are observed as being higher than those of the bare Si (B-Si) electrode cell. The reversible charge capacities were 1468, 1205, 996, 842, 711, and 642 mAhg^−1^ at 0.2, 0.5, 1, 2, 3, and 5 C (Figure 9b), respectively, and their corresponding rate cyclic test is shown in Figure 9d. The carbon modification of Si possessed a positive effect on the rate performance of electrode, which could be ascribed to the boosted the conductivity, as well as electron transfer through the carbon network.

Figure 10a,b show the EIS spectra of before and after the 10 (de)lithiation processes. The fresh cell with a diameter of semicircle signifies the charge transfer resistance (R_ct_) of the Faradaic current, and the low-frequency linear portion is ascribed to the diffusion of Li^+^ ions. However, the cycled cell showed two semicircles; the first circle is due to the SEI and followed by R_ct_. The R_ct_ values of the B-Si and M-Si samples revealed 22.64 Ω and 13.34 Ω for fresh cells. The cycled cells showed the SEI resistance of 161.88 Ω and 67.60 Ω for the sample cells consisting of B-Si and M-Si, respectively. Thus, the C-modified (M-Si) via PAD process showed greatly improved kinetics of Li-ions, with a much smaller R_ct_ and SEI. Therefore, the Si/SiC/C composite material had a higher electrochemical performance than that without a modified sample.

## 4. Conclusions

In summary, the Si/SiC/C nanocomposite material (M-Si) was successfully synthesized from Si-wafer wastage resources by using the plasma-assisted discharge process. The physicochemical and electrochemical properties were systematically compared without the carbon-modified Si (B-Si) sample. The M-Si sample’s morphology showed two kinds of carbon distributions, namely the wrapping of a graphene-like layer and the enfolding of amorphous nature of C on the Si nanoparticles. The M-Si material as an anode using Li-ion storage exhibited reversible discharge/charge capacities of 563/557 mAhg^−1^ at a rate of 0.1C over 100 and 250 cycles. However, the B-Si sample cell showed lower reversible capacity (383/374 mAhg^−1^) and rate capacities than that of the M-Si electrode cell. Hence, this work proves and led to efficacious wastage management from recycling of industrial wastage of Si resource. In addition, the plasma-assisted discharge process will promote the high-quality large-scale Si-composite anode materials for high-performance Li-ion battery applications.

## Figures and Tables

**Figure 1 nanomaterials-12-00659-f001:**
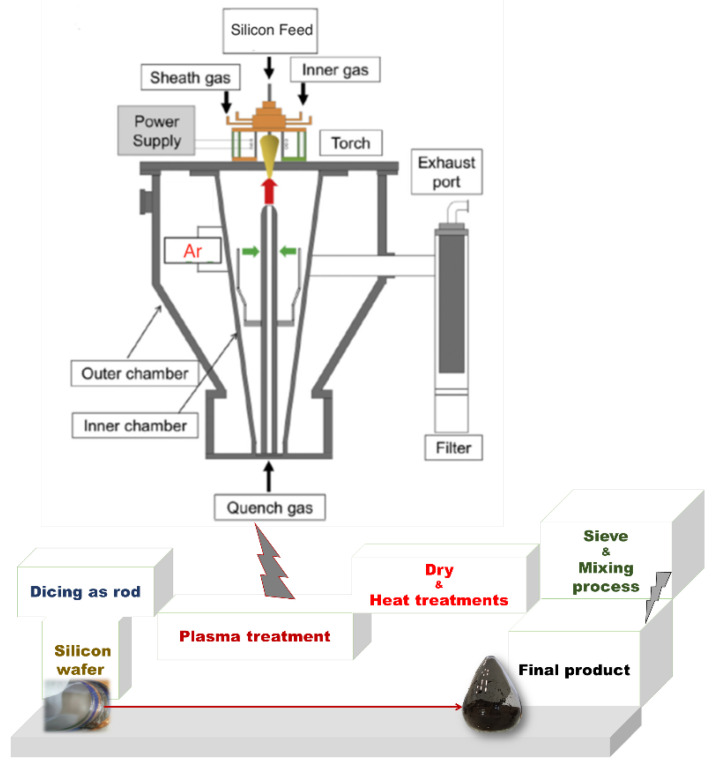
Schematic view of the plasma-assisted discharge method (PAD process).

**Figure 2 nanomaterials-12-00659-f002:**
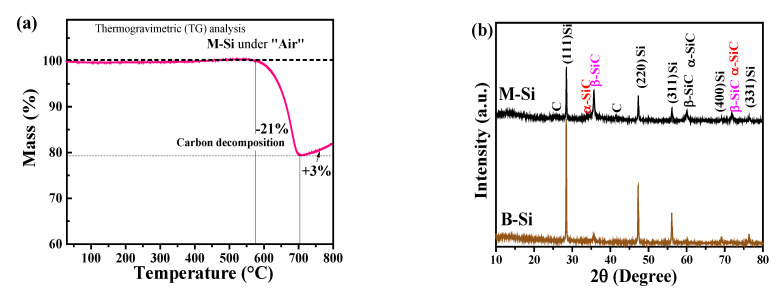
(**a**) TGA curve of M-Si powder in air with a heat rate of 5 °C/min. (**b**) XRD patterns of M-Si and B-Si.

**Figure 3 nanomaterials-12-00659-f003:**
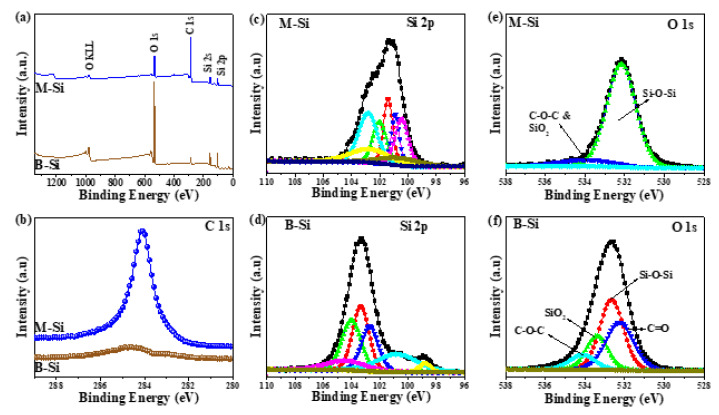
X-ray photoelectron spectra (XPS) of M-Si and B-Si: (**a**) survey scan, (**b**) C 1s, (**c**,**d**) Si 2p, and (**e**,**f**) O1s, respectively.

**Figure 4 nanomaterials-12-00659-f004:**
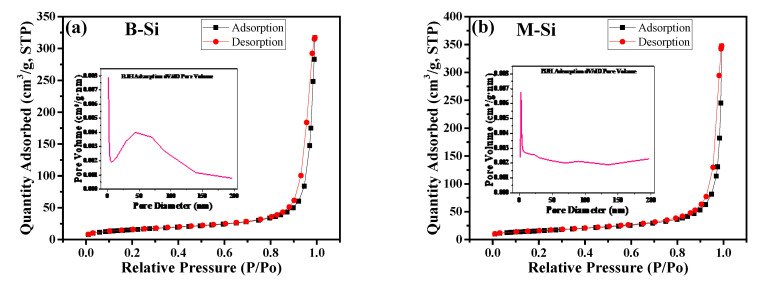
N_2_ adsorption/desorption isotherms of (**a**) B-Si and (**b**) M-Si. Insets: the corresponding pore size distributions of B-Si and M-Si, respectively.

**Figure 5 nanomaterials-12-00659-f005:**
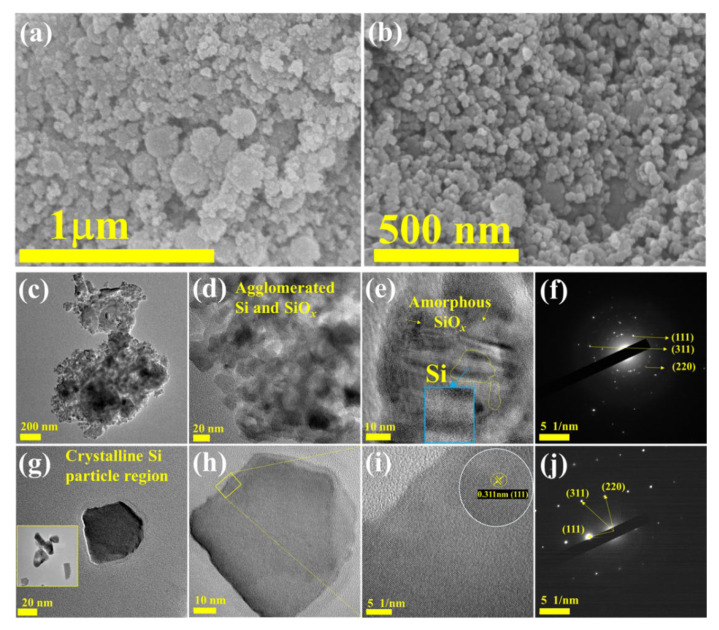
(**a**,**b**) SEM images of B-Si sample with different magnifications. (**c**–**e**) HR-TEM images of agglomerated Si/SiO_x_ regions. (**f**) SAED in region of (**e**). (**g**–**i**) HR-TEM images of crystalline Si-region particle, lattice view, and the corresponding SAED (**j**).

**Figure 6 nanomaterials-12-00659-f006:**
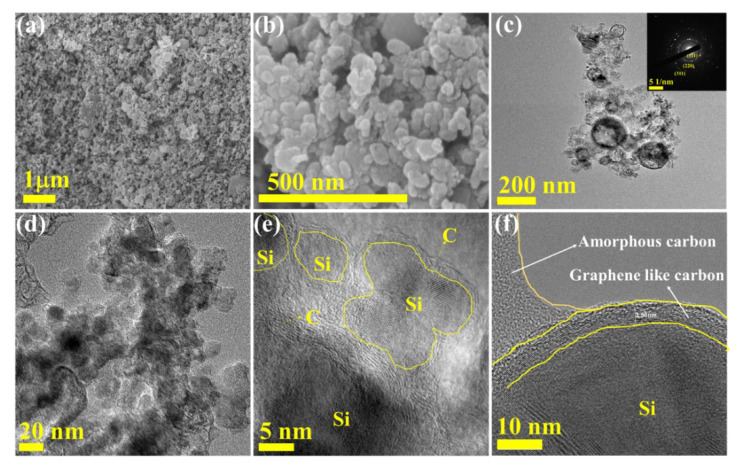
(**a**,**b**) SEM images of M-Si with different magnifications. (**c**–**f**) TEM images of M-Si with different magnitude. Inset in (**c**) is SAED of M-Si.

**Figure 7 nanomaterials-12-00659-f007:**
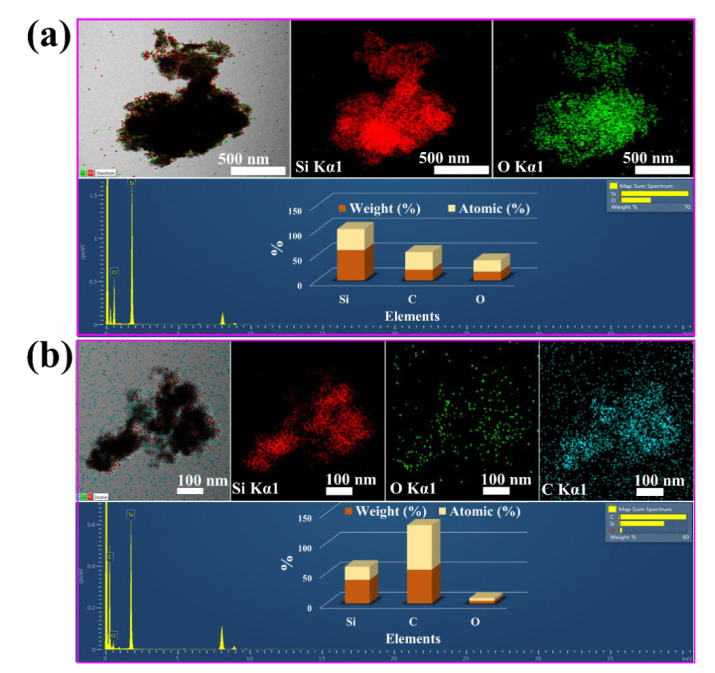
HR-TEM/EDS mapping, corresponding spectrum, and the atomic/weight percentages of (**a**) B-Si and (**b**) M-Si, respectively.

**Figure 8 nanomaterials-12-00659-f008:**
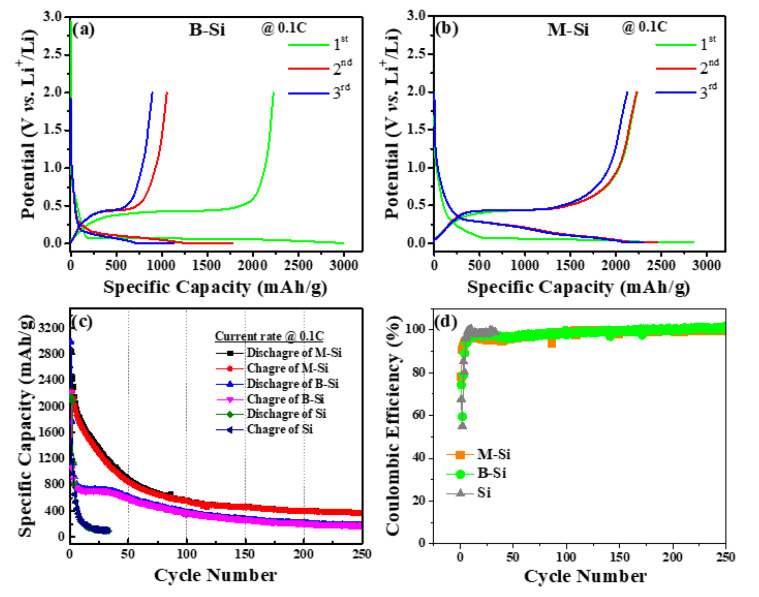
Charge/discharge curves of (**a**) B-Si and (**b**) M-Si at the first three cycle at 0.1C. (**c**) Cycle life tests of pure Si, B-Si, and M-Si at 0.1C. (**d**) Corresponding coulombic efficiency of pure Si, B-Si and M-Si, respectively.

**Figure 9 nanomaterials-12-00659-f009:**
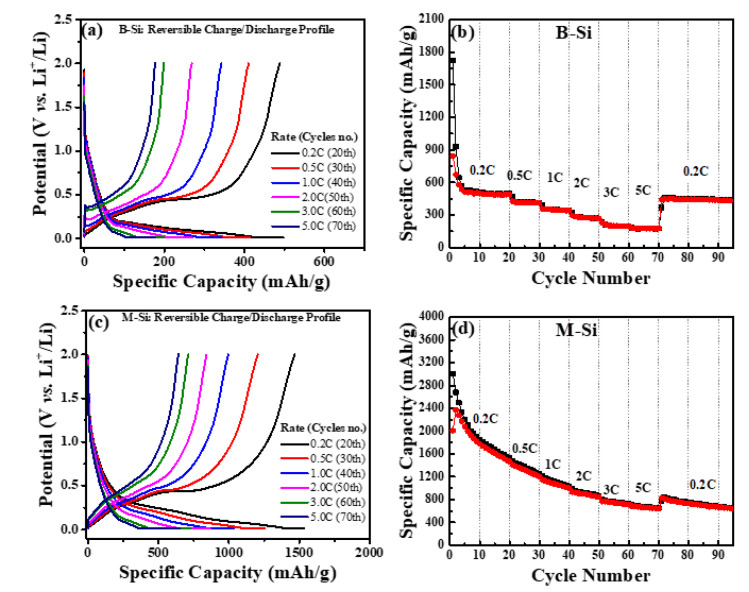
Charge/discharge curves of (**a**) B-Si and (**c**) M-Si electrodes at 20th, 30th, 40th, 50th, 60th, and 70th cycles. Rate capability tests of (**b**) B-Si and (**d**) M-Si electrodes under different current densities of 0.2C, 0.5C, 1C, 2C, 3C, 5C, and 0.2C.

**Figure 10 nanomaterials-12-00659-f010:**
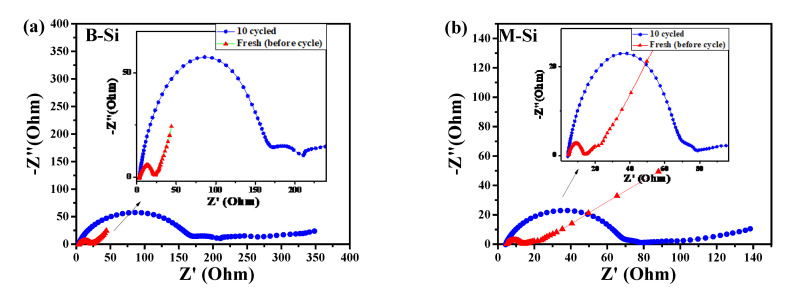
EIS of cells consisting of (**a**) B-Si and (**b**) M-Si material before (fresh) and after 10 cycles (de)lithiation process.

## Data Availability

Not Applicable.

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
