# Peer review of "Industrial Silicon-Wafer-Wastage-Derived Carbon-Enfolded Si/Si-C/C Nanocomposite Anode Material through Plasma-Assisted Discharge Process for Rechargeable Li-Ion Storage"

_nanomaterials, 2022, doi:10.3390/nano12040659_

Round 1

Reviewer 1 Report

The article deals with the modification of Si materials derived from Silicon waster and its electrochemical performances for Lithium ion batteries. The conceptual idea looks reasonable and is scientifically important for the battery society. However, there are couple of things to be improved before the publication. In this regard, I would suggest "minor revision" for this article.  Please check the following commments and address the questions. 

1) The actual mass of active Si materials used for running coin-cells should be given in the main-text. 

2) According to the Figure 8c, the cycle retention of M-Si is poorer than that of B-Si. Basically it is known that the coating layer, i.e., carbon, is capable of suppressing the volume expansion of Si electrode, leading to the improved cycle stability. The result in the Figure 8c is contradict to the previous Si studies. The authors should provide a clear reason for the huge capacity loss of M-Si. 

3) In Figure 2a, TGA result is given. It is highly likely Si would be oxidized when the TGA specimen is heated up to 800 oC under air. This data would not be correct.  Authors should provide other evidence to calculate the mass of Carbon in the Si-C composite. 

4) The authors need to add more relevant papers in the manuscript, in order to make the paper attractive. 

  - Nanomaterials 202111(12), 3248 
  - Scientific Reports volume 5, Article number: 9431 (2015)
  - etc...

1)

Author Response

The article deals with the modification of Si materials derived from Silicon waster and its electrochemical performances for Lithium ion batteries. The conceptual idea looks reasonable and is scientifically important for the battery society. However, there are couple of things to be improved before the publication. In this regard, I would suggest "minor revision" for this article.  Please check the following comments and address the questions. 

  • The actual mass of active Si materials used for running coin-cells should be given in the main-text. 

Responses:

We thank to the reviewer for the professional comment. The active material (bare Si and PAD-modified Si) mass percentage is 70 wt.% and the actual mass of active Si was maintained 1.8–2.0 mg/cm2 for running and fabrications of coin-cells.

We hoped the reviewer could satisfy with our responses for publication.

  • According to the Figure 8c, the cycle retention of M-Si is poorer than that of B-Si. Basically, it is known that the coating layer, i.e., carbon, is capable of suppressing the volume expansion of Si electrode, leading to the improved cycle stability. The result in the Figure 8c is contradict to the previous Si studies. The authors should provide a clear reason for the huge capacity loss of M-Si. 

Responses:   

We thank to the reviewer for the valuable comments. Yes, we agreed with reviewer’s concerns. In fact, the B-Si is derived from Si-wafer and M-Si is modified with carbon. We have included the pure Si nanoparticle (pristine Si) consisted cyclic test in the revised manuscript. It showed poor cyclic stability than as-prepared B-Si and M-Si consisted electrode cells. Thus, the stability of as-synthesized B-Si and M-Si were much higher than that of pristine Si.

        Moreover, the as-prepared M-Si consisted cells showed gradual capacity fading over 50 cycles, which can be suggested from the slowly infiltration of the electrolyte into the electrode porous, continuous formation of further SEI through the improved specific surface area contains with amorphous of carbons and gradual activation process. The similar gradual capacity fading was observed in reported Si/C and Si/C/SiC composites as anode materials for Li-ion batteries. [Nanomaterials 2021, 11, 302.   Scientific Reports 2019, 9, 14814; Journal of Energy Chemistry 2014, 23, 315-323; Composites Part B: Engineering 2021, 215, 108799; RSC Advances 2018, 8, 5189-5196; RSC Advances 2018, 8, 27580-27586.]

For your visibility, the following several reports showed capacity fading over 50 cycles of Si/C and Si/SiC/C composites as anode material for Lithium ion battery.

Tzeng et al. [Nanomaterials 202111(2), 302]

Du et al. [Journal of Energy Chemistry 23(2014)315–323]

In addition, we are further under processing to enhance the better electrochemical performance for high-rate practical Si-anode based Li-ion battery applications through the optimizing of different kinds of composites and other parameters variations. Thus, we will focus on more in-depth studies and discuss of the reviewer concerns in future.

We hoped the reviewer could satisfy with our responses for publication.

  • In Figure 2a, TGA result is given. It is highly likely Si would be oxidized when the TGA specimen is heated up to 800oC under air. This data would not be correct. Authors should provide other evidence to calculate the mass of Carbon in the Si-C composite. 

Responses:

We thank to the reviewer for the professional comment. The following several reports have also been estimated carbon content in Si/C composite through TGA measurements. By the way, we have predicted through the TGA test to estimate the C content in the Si-C composite.

For your reference,

Yang et al. [Yang, L., Li, H., Liu, J. et al. Dual yolk-shell structure of carbon and silica-coated silicon for high-performance lithium-ion batteries. Sci Rep 5, 10908 (2015).] reported the weight ratio of silicon in the Si/void/SiO2/void/C nanocomposite about 64% and that of silica was about 22%.

TGA and DSC curves for Si/void/SiO2/void/C composites.

Zhu et al. [RSC Adv., 2013,3, 6141-6146] reported the content of Si in the Si–Gr composite from the largest weight loss at 22 wt%; that is, the Si fraction is 78 wt% through TGA.

Xu et al. [J. Mater. Chem. A, 2014, 2, 9751-9757] determined the carbon content in the mesoporous C/Si composite using thermogravimetric analysis (TGA) in air. Also, they were calculated Si content in the composite about 76% by weight.

Thermogravimetric analysis (TGA) curve of the mesoporous C/Si composites in air.

Cong et al. [Cong, R., Choi, JY., Song, JB. et al. Characteristics and electrochemical performances of silicon/carbon nanofiber/graphene composite films as anode materials for binder-free lithium-ion batteries. Sci Rep 11, 1283 (2021).] confirmed that the Si content in each composite film of Si:CNF/rGO = 3:2, 1:1, and 2:3 can reach 71.0 wt%, 64.25 wt%, and 56.9 wt%, respectively. The total content of rGO and CNF were measured by the reduction in total weight.

TGA curves of the Si/rGO, Si:CNF/rGO = 3:2, Si:CNF/rGO = 1:1, and Si:CNF/rGO = 2:3 samples.

Thus, we believe the estimated way is also more sensible and acceptable.

We hoped the reviewer could satisfy with our responses for publication.

  • The authors need to add more relevant papers in the manuscript, in order to make the paper attractive.

  - Nanomaterials 202111(12), 3248 
  - Scientific Reports volume 5, Article number: 9431 (2015)
  - etc...

Responses:

We thank to the reviewer for suggesting references and the suggested/relevant references have been included in the revised manuscript.

10. Jang, H.D.; Kim, H.; Chang, H.; Kim, J.; Roh, K.M.; Choi, J.-H.; Cho, B.-G.; Park, E.; Kim, H.; Luo, J., et al. Aerosol-Assisted Extraction of Silicon Nanoparticles from Wafer Slicing Waste for Lithium Ion Batteries. Scientific Reports 2015, 5, 9431.

28. Kim, J.; Kwon, J.; Kim, M.J.; O, M.J.; Jung, D.S.; Roh, K.C.; Jang, J.; Kim, P.J.; Choi, J. A Strategic Approach to Use Upcycled Si Nanomaterials for Stable Operation of Lithium-Ion Batteries. Nanomaterials 2021, 11, 3248.

29. Tzeng, Y.; He, J.-L.; Jhan, C.-Y.; Wu, Y.-H. Effects of SiC and Resorcinol–Formaldehyde (RF) Carbon Coatings on Silicon-Flake-Based Anode of Lithium Ion Battery. Nanomaterials 2021, 11, 302.

36. Andersen, H.F.; Foss, C.E.L.; Voje, J.; Tronstad, R.; Mokkelbost, T.; Vullum, P.E.; Ulvestad, A.; Kirkengen, M.; Mæhlen, J.P. Silicon-Carbon composite anodes from industrial battery grade silicon. Scientific Reports 2019, 9, 14814.

37. Du, Y.; Hou, M.; Zhou, D.; Wang, Y.; Wang, C.; Xia, Y. Interconnected sandwich structure carbon/Si-SiO2/carbon nanospheres composite as high-performance anode material for lithium-ion batteries. Journal of Energy Chemistry 2014, 23, 315-323.

38. Cho, M.K.; You, S.J.; Woo, J.G.; An, J.-C.; Kang, S.; Lee, H.-W.; Kim, J.H.; Yang, C.-M.; Kim, Y.J. Anomalous Si-based composite anode design by densification and coating strategies for practical applications in Li-ion batteries. Composites Part B: Engineering 2021, 215, 108799.

39. Huang, X.D.; Zhang, F.; Gan, X.F.; Huang, Q.A.; Yang, J.Z.; Lai, P.T.; Tang, W.M. Electrochemical characteristics of amorphous silicon carbide film as a lithium-ion battery anode. RSC Advances 2018, 8, 5189-5196.

40.   Xiao, C.; He, P.; Ren, J.; Yue, M.; Huang, Y.; He, X. Walnut-structure Si–G/C materials with high coulombic efficiency for long-life lithium ion batteries. RSC Advances 2018, 8, 27580-27586.

Reviewer 2 Report

The topic itself is interesting and could be of interest to the readership of the journal. However, the paper in its present form is still rough. There are many typos and grammatical omissions.   English erroneous:  

p1 line 42: capacitance --> capacity

p1 line 44: .. such as graphite, lithium and silicon --> ..such as lithium and silicon  

p2 line 52: I can't understand the phrase: ..rapidly capacity decay during the cycle performance. --> rapid capacity fade  

p2 line 56: grammar: ...have been promoted ... --> promote  

p2 line 75: grammar: The PAD method can be successfully produced industrial level high quality large-scale Si based composite anode material.  

p2 lines 78-80: grammar: The PAD-modified Si (M-Si) and commercial bare nano Si (B-Si) were systematically compared the physical, chemical and  electrochemical characterizations through the XRD, SEM, EDX, XPS, BET as well as electrochemical measurements in terms of C/D tests, rate capability and EIS analyses.  

p2. line 84: displayed (past tense) --> shows (change to present tense).

Also, please do not use past tense when referring to figures  

p3 line 111: Point 2.3 (Fabrication of Lithium-ion storage cell) is incorrect. This is not Lithium-ion storage cell. This is half cell (vs. Li) as stated in line 112. Full cells were not assembled and tested in this work.  

p3 line 112: Li is not a reference, but a counter electrode in this case.  

p4. line 118: ... by Ar-filled glove box --> in an Ar-filled glove box  

p4. line 126: ... all electrochemical performances were conducted at RT. --> ... all electrochemical measurements were conducted at RT. etc... 

Please, correct these too numerous grammatical and typo issues.  

Aside from the extensive English corrections necessary before publication, there is another issue with the method itself. Plasma assisted process does not seem to be energy efficient and easily scalable. Please, evaluate the energy consumption for the production of 1 kg of this active material  (in units kWh/kg). Also, the production rate of this process should be very small.

Please, also evaluate the rate of production in e.g. grams/hour. After answering these questions, the last sentence of the Conclusion section,  mentioning that .. 'the plasma-assisted discharge process will promote and the high-quality large-scale Si-composite anode materials' should be better removed from the final version of the manuscript.

Author Response

Comments:

The topic itself is interesting and could be of interest to the readership of the journal. However, the paper in its present form is still rough. There are many typos and grammatical omissions.  English erroneous:

  • p1 line 42: capacitance --> capacity
  • p1 line 44: .. such as graphite, lithium and silicon --> ..such as lithium and silicon  
  • p2 line 52: I can't understand the phrase: ..rapidly capacity decay during the cycle performance. --> rapid capacity fade  
  • p2 line 56: grammar: ...have been promoted ... --> promote  
  • p2 line 75: grammar: The PAD method can be successfully produced industrial level high quality large-scale Si based composite anode material.  
  • p2 lines 78-80: grammar: The PAD-modified Si (M-Si) and commercial bare nano Si (B-Si) were systematically compared the physical, chemical and electrochemical characterizations through the XRD, SEM, EDX, XPS, BET as well as electrochemical measurements in terms of C/D tests, rate capability and EIS analyses.  
  • line 84: displayed (past tense) --> shows (change to present tense).
  • Also, please do not use past tense when referring to figures  
  • p3 line 111: Point 2.3 (Fabrication of Lithium-ion storage cell) is incorrect. This is not Lithium-ion storage cell. This is half cell (vs. Li) as stated in line 112. Full cells were not assembled and tested in this work.  
  • p3 line 112: Li is not a reference, but a counter electrode in this case.  
  • line 118: ... by Ar-filled glove box --> in an Ar-filled glove box  
  • line 126: ... all electrochemical performances were conducted at RT. --> ... all electrochemical measurements were conducted at RT. etc... 
  • Please, correct these too numerous grammatical and typo issues.

Responses:

We thank to the reviewer for mentioned our mistakes and time considerations. In the revised version, we have corrected all those mistakes mentioned by the reviewer and presented in blue color highlights in the revised manuscript. Also, we have checked and corrected carefully in the whole manuscript.

We are sorry for those mistakes and we hope the reviewer could satisfy with the revised version of the manuscript for publication.

Comments:

Aside from the extensive English corrections necessary before publication, there is another issue with the method itself. Plasma assisted process does not seem to be energy efficient and easily scalable. Please, evaluate the energy consumption for the production of 1 kg of this active material (in units kWh/kg). Also, the production rate of this process should be very small. Please, also evaluate the rate of production in e.g. grams/hour. After answering these questions, the last sentence of the Conclusion section, mentioning that .. 'the plasma-assisted discharge process will promote and the high-quality large-scale Si-composite anode materials' should be better removed from the final version of the manuscript.

Responses:

We agree with reviewer’s comments. Indeed, in the laboratory, plasma-assisted process is not energy efficient and easily scalable. However, in this study, the plasma device is home-made and cooperated with company (Sino Applied Technology Co., Ltd). The production rate of M-Si powder is 1000g/hour (250 Kg/month) with 98% yield. The energy consumption for the production of 1 kg of this active material is estimated to be 110 KWh/kg. We hope reviewer could satisfy with our response.

Round 2

Reviewer 2 Report

The authors have changed the manuscript and it can now be published in the Journal Nanomaterials